# Effects of Titanium–Silica Oxide on Degradation Behavior and Antimicrobial Activity of Poly (Lactic Acid) Composites

**DOI:** 10.3390/polym14163310

**Published:** 2022-08-14

**Authors:** Arpaporn Teamsinsungvon, Chaiwat Ruksakulpiwat, Yupaporn Ruksakulpiwat

**Affiliations:** 1School of Polymer Engineering, Institute of Engineering, Suranaree University of Technology, Nakhon Ratchasima 30000, Thailand; 2Center of Excellence on Petrochemical and Materials Technology, Chulalongkorn University, Bangkok 10330, Thailand; 3Research Center for Biocomposite Materials for Medical Industry and Agricultural and Food Industry, Nakhon Ratchasima 30000, Thailand

**Keywords:** titanium silicon oxide, hydrolytic degradation, titania, silica, antimicrobial activity, photocatalytic degradation

## Abstract

A mixed oxide of titania–silica oxides (Ti_x_Si_y_ oxides) was successfully prepared via the sol–gel technique from our previous work. The use of Ti_x_Si_y_ oxides to improve the mechanical properties, photocatalytic efficiency, antibacterial property, permeability tests, and biodegradability of polylactic acid (PLA) was demonstrated in this study. The influence of different types and contents of Ti_x_Si_y_ oxides on crystallization behavior, mechanical properties, thermal properties, and morphological properties was presented. In addition, the effect of using Ti_x_Si_y_ oxides as a filler in PLA composites on these properties was compared with the use of titanium dioxide (TiO_2_), silicon dioxide (SiO_2_), and TiO_2_SiO_2_. Among the prepared biocomposite films, the PLA/Ti_x_Si_y_ films showed an improvement in the tensile strength and Young’s modulus (up to 5% and 31%, respectively) in comparison to neat PLA films. Photocatalytic efficiency to degrade methylene blue (MB), hydrolytic degradation, and in vitro degradation of PLA are significantly improved with the addition of Ti_x_Si_y_ oxides. Furthermore, PLA with the addition of Ti_x_Si_y_ oxides exhibited an excellent antibacterial effect on Gram-negative bacteria (*Escherichia coli* or *E. coli*) and Gram-positive bacteria (*Staphylococcus aureus* or *S. aureus*), indicating the improved antimicrobial effectiveness of PLA composites. Importantly, up to 5% Ti_x_Si_y_ loading could promote more PLA degradation via the water absorption ability of mixed oxides. According to the research results, the PLA composite films produced with Ti_x_Si_y_ oxide were transparent, capable of screening UV radiation, and exhibited superior antibacterial efficacy, making them an excellent food packaging material.

## 1. Introduction

Higher usage rates of plastics all over the world are causing an increasing rate of disposal of this petroleum-based product. In addition, the limited obtainability of petrochemical resources has become a major global concern [1,2]. Poly (lactic acid) (PLA) is one of the most important biocompatible and biodegradable polymers, and it is a sustainable alternative to petrochemical-derived products [3]. PLA is a synthetic biodegradable polymer, made up of a repeated monomer unit: lactic acid (LA). It is derived from renewable and degradable resources such as corn and rice, and decomposes through simple hydrolysis into water and carbon dioxide. PLA has been viewed as one of the most promising materials because of its excellent biodegradability, biocompatibility, composability, renewability, transparency, high strength, and high modulus [4,5]. Moreover, PLA degradation products are non-toxic (at a lower composition) making it a natural choice for biomedical applications [5,6]. Therefore, this polymer has attracted a wide range of attention in various applications. However, PLA has a slow degradation rate and hydrophobicity, so it does not decompose fast enough for industrial decomposers [7,8,9].

To overcome such shortcoming, extra materials are added to fulfill the essential properties such as improved mechanical, heat stability, and barrier properties, and controlled degradation. In addition, the critical hydrolytic degradation rate often limits its application. The rate of degradation could be controlled by adding plasticizers and additives [10]. Although PLA can be degraded with microorganisms in the ground, it takes over two months to decompose, and does not degrade in air [11]. One method to solve this drawback is to add photodegradability filler into PLA, which helps improve degradability under any conditions [12].

Numerous modifications such as copolymerization, plasticization, polymer blends, and polymer composites have been applied to improve some PLA properties. An example of such modifications is to incorporate nanoparticles into the PLA matrix to enhance PLA properties and control the degradation process in various media [13]. The addition of selected nanofillers into PLA, such as organomodified layered silicates (OMLS) [14], carbon nanotubes (CNTs) [15], zinc oxide [16], silica nanofillers (SiO_2_) [17,18], and titanium dioxide (TiO_2_) [19], can enhance PLA’s characteristic features. However, among the nanofillers mentioned above, TiO_2_ is the most widely used. TiO_2_ or titania is well recognized as a valuable material used in applications such as paints or filler in paper, polymers, textiles, photocatalysis, etc. Nano-TiO_2_ particles hold many good properties such as good chemical resistance, high chemical stability [20], attractive photocatalytic activity, excellent photostability, biocompatibility, and antimicrobial activity [21]. Another most-popular nanofiller is silicon dioxide (SiO_2_) or silica, a chemical compound that contains oxygen and silicon. Within inorganic oxide fillers, SiO_2_ admits much concentration because of its well-defined ordered structure, the easy surface modification, high surface area, and cost-effective production [22]. SiO_2_ helps improve the strength, modulus of elasticity, wear resistance, heat and fire resistances, and insulation of properties of polymer materials [23]. Moreover, SiO_2_ has been widely applied in food additives, drug delivery, bioimaging, gene delivery, and engineering. Additionally, SiO_2_ is classified by the FDA as a “generally regarded as safe” (GRAS) agent, thus making it an ultimate candidate for biomedical applications [24]. The binary oxides are prepared for many purposes: to expand the chemical properties, to develop specific textural properties, or to produce a particle with a personalized composition that is known to present unique characteristics (large surface areas, thermal stability, etc.) [25].

Furthermore, the antimicrobial activity of PLA is usually obtained by adding several metal particles and metal oxides such as silver (Ag) particles [26], zinc oxide (ZnO) [27,28], titanium dioxide (TiO_2_) [29,30], and magnesium oxide (MgO) [31] as antibacterial agents.

The objective of the present research was to study PLA composites incorporated with Ti_x_Si_y_ oxide of different concentrations in PLA. We have inspected the influence of TiO_2_, SiO_2_, and Ti_x_Si_y_ oxide on the mechanical properties, thermal properties, morphological properties, and degradation of PLA in various media. In addition, the antimicrobial activity of the PLA composite was investigated. Furthermore, the addition of Ti_70_Si_30_ oxides into PLA significantly improved the photocatalytic efficiency of degrading methylene blue (MB), hydrolytic degradation, and the in vitro degradation of PLA. In addition, PLA, with the addition of Ti_70_Si_30_ oxides, exhibited an excellent antibacterial effect on Gram-negative bacteria (*E. coli*) and Gram-positive bacteria (*S. aureus*), indicating the improved antimicrobial effectiveness of PLA composites. As a result, Ti_70_Si_30_ oxide was used to study the influence of the contents of Ti_x_Si_y_ oxides on the mechanical properties, thermal properties, morphological properties, and degradation of PLA in various media, and the antimicrobial activity of PLA.

## 2. Materials and Methods

### 2.1. Materials

Poly (lactic acid) (PLA, grade 4043D) was supplied from Nature Works LLC (Minnetonka, MN, USA), a commercial grade for 3D printing and film applications. Tetraethylorthosilicate (TEOS, 98%, AR grade) and Titanium (IV) isopropoxide (TTIP, 98%, AR grade) were purchased from Acros (Geel, Belgium). Absolute ethanol (C_2_H_5_OH, AR grade), hydrochloric acid (HCl, AR grade), and ammonium hydroxide (NH_4_OH, AR grade) were supplied from Carlo Erba Reagents (Emmendingen, Germany). TiO_2_, SiO_2_, and Ti_x_Si_y_ oxide prepared in-house were used as filler [32,33]. Particle size of SiO_2_, Ti_70_Si_30_, Ti_50_Si_50_, and Ti_40_Si_60_ oxide used in this study were in the range of 130–150 nm. TiO_2_ can be generally synthesized under high acid conditions to obtain a uniform spherical shape and small nanoparticles in the range of 25–50 nm [34]. However, the author could synthesise TixSiy under alkali conditions to control hydrolysis step of Ti-precursor to obtain a uniform shape and size in the range of 130–150 nm because Ti-precursor decreased the reactivity of the alkoxide, hence low concentration of Ti-precursor decreased reactivity of the alkoxide with the lower hydrolysis rate [35].

### 2.2. Preparation of PLA Composite Films

PLA and PLA composites were prepared by solvent film casting method. First, Ti_70_Si_30_, Ti_50_Si_50_, Ti_40_Si_60_, SiO_2_, and TiO_2_ were dispersed in chloroform with ultrasonic treatment for 1 day. After that, PLA was added to the Ti_x_Si_y_ oxide and strictly stirred for 4 days. The dispersions of PLA composites were additionally ultrasonically treated for 1 h with frequency of 42 kHz for four times per day. Then, the treated dispersions were slightly poured onto Petri dishes, and the solvent was evaporated under room temperature. The films were dried to constant mass at room temperature for ~24 h and stored in oven at 40 °C for 4 h. Pure PLA and PLA composite films had uniform thickness of 250 ± 4.68 µm. After that, Ti_x_Si_y_ oxides were synthesized using a modified Stöber method involving simultaneous hydrolysis and condensation of TEOS, and Ti_x_Si_y_ mixed oxides’ preparation was reported in details in Teamsinsungvon et al. [32].

### 2.3. Mechanical Properties

The tensile properties of PLA and PLA composites films were obtained in accordance with the ASTM standard method D882-18 using an Instron universal testing machine (UTM, model 5565, Norwood, MA, USA) with a load cell of 5 kN. Specimen samples were 10 cm × 2.54 cm. Crosshead speed was set at 50 cm/min. The values were presented as the average of seven measurements.

### 2.4. Thermal Properties

Thermal properties of PLA and PLA composites films were carried out using a differential scanning calorimeter (DSC204F1, Netzsch, Selb, Germany) equipped with a liquid nitrogen cooling system. The samples were heated from room temperature to 180 °C with a heating rate of 5 °C/min (1st heating scan) and stored for 5 min to erase previous thermal history. Then it was cooled to room temperature (25 °C) with a cooling rate of 5 °C/min. Finally, it was heated again to 180 °C with heating rate 5 °C/min (2nd heating scan). The degree of crystallinity (*X_c_*) of PLA and PLA composites was estimated using Equation (1) [36,37]:(1)$$X_{c}=\frac{\Delta H_{m}}{\Delta H_{m}^{0}\;\cdot \left({Ø}_{PLA}\right)}\;\cdot \;100\;$$
where $\Delta H_{m}^{}$ are the melting enthalpy in the second heating process, $\Delta H_{m}^{0}$, which is the melting enthalpy of an infinitely large crystal, was taken as 93.6 J/g [38], and Ø_PLA_ is the PLA weight fraction in the composites.

Thermogravimetric analysis of PLA and PLA composite films were examined using thermogravimetric analyzer (TGA/DSC1, Mettler Toledo, Columbus, OH, USA). The temperature was raised from the room temperature to 650 °C under nitrogen and then heated to 800 °C under air atmosphere at heating rate of 10 °C/min. The weight change was recorded as a function of temperature.

### 2.5. Morphological Properties

Morphological properties of the film were examined through a scanning electron microscope (JEOL, JSM-6010LV, Tokyo, Japan). An EDAX Genesis 2000 energy dispersive spectrometer (AMETEK, Inc., Berwyn, PA, USA) was applied to determine the spatial distribution of Ti and Si in PLA composite. SEM images of the samples were collected using acceleration voltage 9–12 kV. The cross-sections of the films after tensile test and the freeze-fractured films in liquid nitrogen were sputtered with gold.

### 2.6. Water Vapor Transmission Rate (WVTR)

WVTR, a modified ASTM standard method E96, is a measure of the rate of water that passes through PLA and its composites film at a particular time interval. To analyze, the composite films were cut into circular discs of 1.5 cm diameter and placed on the top of the glass vial containing 5 mL water. Constant humidity was maintained by placing it in the desiccators maintained at room temperature and relative vapor pressure (RVP) = 0 by using silica gel. The vial assembly was weighted every hour on first day, and then the weighting was performed daily over a 20-day period. In terms of statistical approach, when the straight line adequately fit the weight change vs. time plot using linear regression with r^2^ ≥ 0.99, the constant rate of weight change was obtained. *WVTR* was calculated using the Equation (2) [39,40]. *WVTR* (g m^−^^2^ day^−^^1^) for each type of film was determined using three individually prepared films as the replicated experimental units.
(2)$$WVTR=\frac{G\;}{t\;\cdot \;A}\;$$
where *G* = weight change of the vial with water and film (from the straight line) (g), *T* = the duration for the measurement (day), *G/t* = slope of the straight line (g day^−^^1^), and *A* = the test area of the film (m^2^).

### 2.7. Photocatalytic Degradation of Methylene Blue (MB)

The photocatalytic activity of PLA composite films was evaluated by degrading methylene blue (MB) under UV light according to Chinese standard GB/T 23762-2009. Specimens of pure PLA and PLA composite films were 5 cm × 15 cm. Five pre-wetted film of each pure PLA and PLA composite films were placed in a flask, then 200 mL of MB solution (10 mg/L) was added. The flask was placed on a mechanical shaker at 50 rpm in a UV chamber with 4 UV lamps (LP Hg lamps, 8 watts, main light emission at 245 nm). The 4 mL MB solution was collected every 60 min, and analyzed using UV-vis spectrophotometer (Cary300, Agilent Technology, Santa Clara, CA, USA). To maintain the volume of solution in the flask, the samples were placed back after each measurement. The maximum absorbance of MB occurs at 664 nm (Figure 1a). The spectrometer was calibrated with solution of MB at 1 mg/L, 3 mg/L, 5 mg/L, 7 mg/L, and 10 mg/L concentrations, respectively. Calibration curve of methylene blue aqueous solutions are shown in Figure 1b. The samples and the solution were stored in a black box to correct the possible decompositions of MB under UV light in absence of any photocatalyst. The concentration of MB was also measured every 60 min to evaluate the absorption of MB. The values were presented as the average of five measurements.

### 2.8. Light Transmittance and Opacity Measurements

The UV-vis transmission spectra of the film specimens were recorded in the range of 200–800 nm using a UV-vis spectrophotometer (Cary300, Agilent Technology, Santa Clara, CA, USA), according to a modified standard procedure of the British Standards Institution (BSI 1968). The film specimens were directly placed in cells, and empty cells were used as reference. The opacity of the film specimens was determined with well-controlled thicknesses. Equation (3) was used to calculate the *Opacity* (AU.nm∙mm^−^^1^) of the films:(3)$$Opacity=\frac{Abs_{600}}{b}\;\;$$
where *Abs*_600_ = the absorbance at 600 nm; the transmittance values were converted to absorbance values using the Lambert–Beer equation and *b* = the film thickness (mm). This test was triplicated for each type of film [41].

### 2.9. Hydrolytic Degradation

Hydrolytic degradation of PLA and PLA composite films (10 *×* 10 mm^2^) were carried out at 37 °C in small bottles containing 1 mol·dm^−^^3^ NaOH solutions (pH 13). Following the incubation for a given time (0, 60, 120, 180, 240, 300,360, and 420 min), the films were periodically removed, washed with distilled water, and dried in oven at 40 °C for 48 h. The weight loss (*W_loss_*) was estimated using following Equation (4):(4)$$W_{loss}\left(\%\right)=\frac{W_{0}-W_{t}}{W_{0}}\cdot 100\;$$
where *W*_0_ = the initial weight of polymer film, and *W_t_* = the weight of degraded sample measured at time *t* after drying in oven for 48 h [12].

### 2.10. In Vitro Degradation

The study of degradation of PLA and its composite was carried out following the standards specified by BS EN ISO 10993-13:2010 [42] and ASTM F1635-11 [43]. Each of PLA and PLA composite samples of (0.4–0.5 g) was accurately weighed. The samples were then immersed separately in 0.01 M phosphate-buffered saline (PBS) (pH = 7.4 ± 0.2) solution and maintained at 37 °C with different soaking times from 1 to 8 weeks (0, 2, 4, 6, 8). At various time points, the specimens were washed with deionized water to remove the salts, then oven dried at 40 °C for 48 h. Later, dry weights of the samples were recorded. The percentage mass change was determined using the following Equation (5);
(5)$$W_{loss}\left(\%\right)=\frac{W_{t}-W_{0}}{W_{0}}\cdot 100\;$$
where *W_t_* = mass of degraded sample measured at time *t* after drying at 40 °C in oven for 48 h, and *W*_0_ = the initial mass of the sample. At each time point, every sample was weighed and mechanically tested, allowing the degradation pathway of each individual sample to be followed with time.

The tensile properties of each specimen were measured by an Instron universal testing machine (UTM, model 5565) with a load cell of 5 kN and crosshead speed of 50 cm/min. The values were presented as the average of five measurements.

### 2.11. Antimicrobial Activity

Antimicrobial effects of the different samples were determined using the JIS Z 2801:2006 method. The ability of PLA and PLA composite films to restrain the growth of Escherichia coli and Staphylococcus aureus were investigated. The bacteria were incubated at 37 °C for 24 h. A plate containing a test sample was inoculated with 0.2 mL of an overnight culture of Escherichia coli and Staphylococcus aureus, while bacterial culture concentration was adjusted to 10^6^ CFU/mL. All petri dishes were incubated at 37 °C for 24 h and colony-forming units (CFU) were counted. Percentage reduction of the colonies was calculated using Equations (6) and (7) below, which relates the number of colonies of neat PLA with that of the composites.
(6)$$\%\;Reduction=\frac{\left(Log\;CFU\;at\;0\;h-Log\;CFU\;at\;24\;h\;\right)}{Log\;CFU\;at\;0\;h}\;\cdot 100\;$$
(7)Antimicrobial activity(R)=Ut−At
where *Ut* = average of CFU per milliliter after inoculation on untreated test pieces after 24 h; *At* = average of CFU per milliliter after inoculation on antibacterial test pieces after 24 h.

## 3. Results and Discussion

### 3.1. Mechanical Properties

The tensile properties of PLA and PLA composites with various Ti_70_Si_30_ oxide contents and different types of nanoparticles are listed in Table 1.

The addition of 3 wt.% of Ti_70_Si_30_ and Ti_50_Si_50_ slightly increased the tensile strength and Young’s modulus and decreased the elongation at break of the PLA composite films compared to neat PLA (Figure 2). This may be due to higher interfacial adhesion between the oxide filler and the PLA matrix by the Van der Waals force or induction interactions, which decreased thereafter when adding a mixed oxide content of up to 5 wt.%, however, it was slightly higher than pure PLA. The result could be attributed to the increased filler quantity leading to a weaker filler–matrix interface and the agglomeration of filler particles, which consequently decreases the tensile strength. The Young’s modulus of the composites insignificantly varied in correspondence with Ti_x_Si_y_ oxide (Figure 2c). However, elongation at the break decreased with the addition of Ti_x_Si_y_ oxide (Figure 2b) as a result of the addition of a rigid phase in the PLA composite, which contributed to a reduction of the PLA ductility. Nevertheless, the addition of TiO_2_SiO_2_ and Ti_40_Si_60_ in the PLA matrix increased elongation at the break of PLA.

During tensile testing, it was observed that the fracture behavior of the film changed for PLA, PLA/TiO_2_, PLA/SiO_2_, PLA/Ti_x_Si_y_, and PLA/TiO_2_SiO_2_ composites. This was demonstrated in the tensile stress–strain curves, as shown in Figure 3. With the addition of filler to the PLA matrix, the composite exhibits elastic behavior, enhancing the toughness of PLA. Furthermore, it was shown that the addition of Ti_70_Si_30_ oxide to the PLA polymer matrix resulted in a decrease in the ductile characteristics, while slightly increasing the tensile strength and decreasing elongation at the break. However, the addition of TiO_2_SiO_2_ and Ti_40_Si_60_ oxide increased elongation at the break of PLA. This may be due to the particles of TiO_2_SiO_2_ in the formulation being homogenously distributed in the polymer matrix, contributing to the occurrence of plastic deformations in the whole sample, allowing a higher elongation than that of the pure PLA.

### 3.2. Thermal Properties

Thermal behaviors of pure PLA and PLA composites with TiO_2_, SiO_2_, Ti_70_Si_30_, Ti_50_Si_50_, Ti_40_Si_60_, and TiO_2_SiO_2_ were investigated using differential scanning calorimetry (DSC), in which first heating scans (1st heating scans), cooling scans, and second heating scans (2nd heating scans) of PLA and PLA composites were performed. DSC thermograms of PLA and PLA composites are shown in Figure 4. The thermal properties of all samples are listed in Table 2. The glass transition temperature (T_g_), cold crystallization temperature (T_cc_), and melting temperature (T_m_) of the PLA composites were observed.

T_g_ and T_cc_ of neat PLA occur at 58.87 °C and 109.47 °C, respectively, while T_m_ appeared at 145.84 °C and 152.35 °C, respectively. The double melting endotherms of neat PLA were explained by the melting and recrystallization. The peak at a low temperature was attributed to the melting of the crystals formed during the non-isothermal melt crystallization, while the peak at a high temperature corresponded to the re-melting of the newly-formed crystallite during melting, and recrystallization during the DSC heating scans [44]. However, T_c_ of the neat PLA did not appear in the cooling cycle. The addition of TiO_2_, SiO_2_, Ti_70_Si_30_, Ti_50_Si_50_, Ti_40_Si_60_, and TiO_2_SiO_2_ in the PLA matrix showed insignificant effects on the glass transition temp (T_g_), cold crystallization temp (T_cc_), and melting temp (T_m_) of PLA. After the cooling scan (Figure 4b), no crystallization temperatures (T_c_) were observed in all composites in the cooling cycle. However, by adding 5wt.% of Ti_70_Si_30_ oxide into PLA, T_c_ was only observed at 101 °C when cooling the sample in the DSC measurement at a cooling rate of 5 °C/min.

Cold crystallization is a phenomena that occurs from re-crystallization during the heating process of polymers [45]. Cold crystallization behaviors (Figure 4c) of PLA films containing 3 wt.% of TiO_2_, SiO_2_, Ti_70_Si_30_, Ti_50_Si_50_, Ti_40_Si_60_, and TiO_2_SiO_2_ nanoparticles and 5 wt.% of Ti_70_Si_30_ were observed. For neat PLA, there is a slightly cold crystallization peak around 109.47 °C. While adding 3 wt.% of TiO_2_, SiO_2_, Ti_70_Si_30_, Ti_50_Si_50_, Ti_40_Si_60_, and TiO_2_SiO_2_ into PLA, the cold crystallization peak of PLA composites shifted to a lower temperature by approximately 1–2 °C. Moreover, the T_cc_ of the PLA composite shifted to 103.09 °C with Ti_70_Si_30_-loading rising to 5 wt.%, showing that a 5 wt.% Ti_70_Si_30_ addition can promote PLA crystallization. This might be attributed to the increase of the chain mobility of PLA, and the Ti_x_Si_y_ oxide could act as an efficient cold crystal nuclei site, which consequently increased the crystallinity of PLA. These results suggest that Ti_70_Si_30_ oxide had a positive effect on the promotion of the crystallization of PLA and could act as a nucleating agent. Similar to other results, Chen et al. [45] found that by incorporating a composite nucleating agent (CNA) to PLA, the polymer could decrease T_cc_, indicating that the crystallization ability of the PLA composite can be enhanced in such a way.

The addition of 5 wt.% Ti_70_Si_30_ is found to be able to enhance the T_m_ of PLA composites. This is possibly a result of the heterogeneous nucleation effects of Ti_70_Si_30_ nanoparticles on PLA during the crystallization process. The lamella formation of PLA was hindered by Ti_70_Si_30_ and led to less perfect crystals of PLA [46]. While the addition of a 3 wt.% of TiO_2_, SiO_2_, Ti_70_Si_30_, Ti_50_Si_50_, Ti_40_Si_60_, and TiO_2_SiO_2_ did not significantly affect the T_m_ of PLA, the degree of crystallinity (χ_c_) of neat PLA significantly increased with incorporating TiO_2_, SiO_2_, Ti_70_Si_30_, Ti_50_Si_50_, Ti_40_Si_60_, and TiO_2_SiO_2_, indicating that TiO_2_, SiO_2_, Ti_70_Si_30_, Ti_50_Si_50_, Ti_40_Si_60_, and TiO_2_SiO_2_ can act as nucleating agents for PLA.

Thermal degradation at 5% weight loss (T_0_._05_), 50% weight loss (T_0_._5_)_,_ final degradation (T_f_), and the char formation at 800 °C of PLA and PLA composites are listed in Table 3, respectively. TGA and DTG curves of PLA and PLA composites at a heating rate of 10 °C/min are shown in Appendix A, respectively. The presence of TiO_2_, SiO_2_, Ti_70_Si_30_, Ti_50_Si_50_, Ti_40_Si_60_, and TiO_2_SiO_2_ did not change the thermal decomposition behavior of PLA, while the mass loss between 250–365 °C was observed, which corresponded to the decomposition of PLA. Then from 365 to 600 °C thermal analysis curves slowed down to complete the decomposition of the PLA matrix until a constant mass was reached. The constant mass remaining at the end of each TGA experiment corresponded to amounts of nanoparticles in PLA composites. In this study, the temperature at 5% weight loss (T_0_._05_) was defined as the onset degradation temperature for the evaluation of the TiO_2_, SiO_2_, and Ti_x_Si_y_ oxide effects on the thermal stability of the PLA composites.

It is obvious that the T_onset_ of the PLA composites shifted to a higher temperature with the presence of 3 wt.% of TiO_2_, SiO_2_, Ti_50_Si_50_, Ti_40_Si_60_, and TiO_2_SiO_2_. Consequently, the thermal stability of the PLA composites was improved. A possible reason to explain this behavior is that TiO_2_, SiO_2_, Ti_50_Si_50_, Ti_40_Si_60_, and TiO_2_SiO_2_ particles may act as a heat barrier in the early stage of thermal decomposition [12]. Similar data have been reported by Zhang et al., who studied PLA composites obtained by adding TiO_2_ to poly (lactic acid) [47]. However, PLA with the addition of 3 wt.% of Ti_70_Si_30_ oxide was found to present a lower onset temperature than that of pure PLA, which resulted in a decrease in the PLA thermal stability. Moreover, it was found that the onset degradation temperature of the composites shifted to a lower temperature with increasing Ti_70_Si_30_ oxide nanoparticles loading from 3 to 5 wt.%. This suggests that there might be degradation due to the water absorption of the filler that would be associated with the cleavage of the chain of PLA at the ester group (–C–O–) by water molecules due to hydrolysis leading to decreased the thermal stability of PLA.

Moreover, the peak in the DTG curves represented the temperature maximum degradation rate (Appendix A). PLA/3TiO_2_ exhibited the fastest degradation rate at the highest temperature, compared to neat PLA and other PLA composites. However, the degradation temperature of the PLA/Ti_70_Si_30_ composite shifted to a lower temperature. This suggested that the thermal stability of PLA decreased with the incorporation of Ti_70_Si_30_ loading. In addition, when 3 wt.% TiO_2_, SiO_2_, Ti_70_Si_30,_ Ti_50_Si_50_, Ti_40_Si_60,_ and TiO_2_SiO_2_, and 5 wt.% Ti_70_Si_30_ mixed oxides were added to PLA, the composites left the char residual of fillers at 4.07, 4.14, 3.91, 4.22, 3.53, 4.50, and 5.46%, respectively, for the PLA composites. The char residual generally depended on the amount of added nanoparticles [48].

### 3.3. Morphological Properties

In order to investigate the dispersion and distribution of TiO_2_, SiO_2_, Ti_70_Si_30_, Ti_50_Si_50_, Ti_40_Si_60_, and TiO_2_SiO_2_ in the PLA composites films, SEM analysis was performed. SEM micrographs of the fracture surface of PLA and PLA adding 3 wt.% of TiO_2_, SiO_2_, Ti_70_Si_30_, Ti_50_Si_50_, Ti_40_Si_60_, and TiO_2_SiO_2_, and 5 wt.% of Ti_70_Si_30_ are shown in Figure 5a, and the surface of the PLA and PLA composites films after the tensile test is shown in Figure 5b.

The SEM results showed that a relatively brittle and comparatively flat surface without holes and air bubbles was found on the fracture surface of the pure PLA films. Meanwhile, SEM images of all PLA composites exhibited roughness caused by adding 3 wt.% of TiO_2_, SiO_2_, Ti_70_Si_30_, Ti_50_Si_50_, Ti_40_Si_60_, and TiO_2_SiO_2_ nanoparticles, particularly at 3 wt.% of TiO_2_SiO_2_ (Figure 5a). The enhancement of the mechanical properties depended on the absence of voids, undamaged position of fillers, interfacial bonding between the fillers and matrix, and the absence of an agglomerate of fillers [49]. However, the white spots in the PLA composites micrographs illustrates the agglomerates of TiO_2_, SiO_2_, Ti_70_Si_30_, Ti_50_Si_50_, Ti_40_Si_60_, and TiO_2_SiO_2_ in the PLA matrix, leading to poor mechanical properties. In this work, although some agglomerations could be observed in all PLA composite films, 3wt.% of Ti_70_Si_30_ and Ti_50_Si_50_ was still kept intact within the PLA matrix (Figure 5b). As Ti_70_Si_30_-loading was increased to 5 wt.%, the position of Ti_70_Si_30_ in PLA was displaced, leading to the formation of a gap between the filler surface and PLA matrix. Therefore, it is an indication of poor interfacial adhesion between Ti_70_Si_30_ and PLA at high loading [50].

The EDX elemental mapping results (Figure 6b–e) suggested the existence of Ti_x_Si_y_ mixed oxides in the PLA composites. Furthermore, EDX elemental analysis results (Figure 6f) of the selected area also confirmed the spatial distribution of the Si, Ti, and O elements of the Ti_x_Si_y_ mixed oxide in the PLA composite. The distribution of Si and Ti in the particles was relatively uniform in the case of the PLA/3Ti_70_Si_30_ composite.

### 3.4. Water Vapor Transmission Rate (WVTR)

One of the most important properties of bio-based composites films is the ability to evaluate the moisture transfer from the environment to the product. The WVTR of the PLA and PLA composite films is shown in Figure 7. The WVTR of the PLA films was 0.316 g m^−2^ day^−1^ which was lower than the PLA films incorporated with 3wt.% of SiO_2_, Ti_70_Si_30_, and Ti_50_Si_50_, which were 1.000, 1.023, and 0.523 g m^−2^ day^−1^. In addition, the WVTR of the PLA/Ti_70_Si_30_ composite film increased with increasing Ti_70_Si_30_ content to 5 wt.%. It is common that, for a solid polymer, the water vapor transmission follows a simple mechanism including adsorbing at the entering face, dissolving, and rapidly creating equilibrium, diffusing through the film, and desorbing at the exit face [51]. The smaller the particle diameter of the nanoparticles is, the more the indirect pathway reducing the diffusion coefficient is produced [52,53]. In other words, the particle diameter is indirectly proportional to the diffusion coefficient. Consequently, the hydrophilicity of the PLA composite incorporating SiO_2_, Ti_70_Si_30_, and Ti_50_Si_50_ was improved.

### 3.5. Photocatalytic Degradation of Methylene Blue (MB)

In this study, the photocatalytic activity of PLA and PLA composite films were investigated by degrading methylene blue (MB). The decomposition of MB might be caused by UV irrigation without the presence of any photocatalyst.

Figure 8 shows changes in the concentration of MB in an aqueous solution under UV irrigation, which was a result of MB decomposition. The presence of 3 wt.% TiO_2_, SiO_2_, Ti_70_Si_30_, Ti_50_Si_50_, Ti_40_Si_60_, and TiO_2_SiO_2_ in the PLA film matrix exhibited MB degradation more efficiently than using photocatalysis solely. The efficiency to degrade MB was TiO_2_ > Ti_70_Si_30_ > TiO_2_SiO_2_ > Ti_50_Si_50_ > Ti_40_Si_60_ > SiO_2_, respectively. It was also found that an increase in Ti_70_Si_30_-loading to 5wt.% improved the efficiency of the photocatalytic activity of PLA. The photo-activity of the mixed oxide was evidently increased because the high content of the mixed oxide’s increasing surface area of the filler effectively concentrated MB around the nanoparticle and produced high concentrations of organic compounds for the photocatalysis, which consequently improved the photocatalytic activity of PLA. It is known that photocatalytic activity occurs at the surface of the photocatalyst. Therefore, the surface area of PLA composites film, which in turn depends on the size of the nanoparticles, film morphology, and thickness, has an effect on photocatalytic reactivity [54]. The PLA composite film containing TiO_2_ can degrade MB more effectively than that containing only photocatalysis. This may be due to two reasons. Firstly, MB was degraded directly by UVC. Secondly, TiO_2_ received light energy more than band-gap energy and then the electron in the valence band (VB) was excited to the conduction band (CB), resulting in a generated hole (*h^+^*) (Equation (8)). This hole could oxidize *MB* (Equation (9)) or oxidized *H*_2_*O* to produce *OH* (Equation (10)). The *e*^−^ in CB could reduce *O*_2_** at the surface of *TiO*_2_** to generate $O_{2}^{-}$ (Equation (11)). The appearance of radical (*OH*, $O_{2}^{-}$) and *h^+^* reacted with *MB* to generate a peroxide derivative and hydroxylate or degrade completely to *CO*_2_** and *H*_2_*O* [54]. The photodegradation mechanism can be summarized by Equations (8)–(11).
(8)$$TiO_{2}+UVC\;\to \;e^{-}+h^{+}\;$$
(9)$$h^{+}+MB\;\to \;CO_{2}+H_{2}O\;$$
(10)$$h^{+}+H_{2}O\;\to \;H^{+}+OH\;$$
(11)$$e^{-}+O_{2}\;\to \;O_{2}^{-}\;$$

The presence of filler in the PLA film matrix shows more efficiency to degrading MB than using only photocatalysis. The enhanced photocatalytic properties of the PLA composites could be mainly attributed to promoted surface adsorption and mass transfer/diffusion, increased light absorption and utilization efficiency, and especially, the higher charge transfer and separation rate by the filler, increased light-harvesting ability, and promoting photoexcitation charge separation, which were the main reasons for improving photocatalytic activity [55,56]. In addition, the photocatalytic activity is influenced by the crystal structure, particle size, specific surface area, and porosity of nanoparticles. So, ultrafine powders of mixed oxide show good catalytic activity. However, agglomeration often takes place, resulting in the reduction or even complete loss of photocatalytic activity.

Due to its photocatalytic activity, Ti_70_Si_30_ nanoparticles with a high specific surface area (569 m^2^ g^−1^) [33] can degrade MB, making it a suitable material for photocatalytic application.

### 3.6. Light Transmittance and Opacity Measurements

UV light can create free radicals in products by a photochemical reaction, leading to a negative effect for food. Some of the unfriendly effects include the deterioration of antioxidants, destruction to vitamins and proteins, and a change in color. UV radiation is classified into UV-A (wavelength 320–400 nm), UVB (280–320 nm), and UV-C (200–280 nm) [57,58]. For good optical properties, the optical transmittance should exceed 90% in the visible range (measured from 400 to 800 nm). So, the optical transmittance was measured using a light source with a 600 nm wavelength, which is the central wavelength of the visible range. So, a lot of research uses this wavelength to evaluate opacity [41,59,60]. The addition of TiO_2_, SiO_2_, Ti_70_Si_30_, Ti_50_Si_50_, Ti_40_Si_60_, and TiO_2_SiO_2_ into the PLA matrix caused a significant decrease of transmittance in all UV regions (Table 4). The results show that the addition of filler into the PLA matrix caused a significantly decrease of transmittance in all UV regions. The presence of 3wt.% Ti_70_Si_30_ in the PLA film matrix succeeded in blocking more than 99.6% of the 240, 300, and 360 nm wavelengths, as representatives of UV-C, UV-B, and UV-A radiation, respectively, with a low opacity for the composite film. Moreover, the increase of Ti_70_Si_30_ oxide-loading up to 5wt.% into PLA improved the UV-blocking efficiency.

The PLA films were transparent and colorless, and the addition of SiO_2_ to PLA remained transparent, while other PLA composite films showed higher opacity than the pure PLA film. However, the transparency changes related to the increasing Ti_70_Si_30_ oxide contents from 3 up to 5 wt.% provided totally opaque films by more than two orders of magnitude in the opacity films, but this was still lower than the composite film of PLA with TiO_2_. Similarly, the addition of TiO_2_ and Ti_70_Si_30_ made the PLA composites’ color appear whiter because of the characteristic whiteness of the TiO_2_ and Ti_70_Si_30_ nanoparticles. Photographs of PLA, PLA/TiO_2_, PLA/SiO_2_, PLA/Ti_x_Si_y_, and PLA/TiO_2_SiO_2_ composites films are shown in Figure 9. These results suggest that the PLA composite produced with Ti_70_Si_30_ oxide was suitably applied to transparency packaging with good UV-blocking efficiency.

### 3.7. Hydrolytic Degradation

Figure 10 shows the percentage weight loss of PLA and PLA composite films as a function of hydrolytic degradation time. Complete degradation of PLA was achieved at about 1200 min, while all of the PLA composite films were hydrolyzed faster than neat PLA. Interestingly, the incorporation of 3 wt.% of TiO_2_, SiO_2_, Ti_70_Si_30_, Ti_50_Si_50_, Ti_40_Si_60_, and TiO_2_SiO_2_ exhibited a much higher weight loss as a function of time than neat PLA. The presence of filler induced a much more apparent change of weight loss of hydrolytic degradation, which indicates the enhancement of a hydrolytic degradation ability of the PLA matrix. This was attributed by the addition of nanoparticles, which helped accelerate the hydrolytic degradation of the PLA matrix. Furthermore, 97PLA/3TiO_2_, 97PLA/3SiO_2_, 97PLA/3Ti_70_Si_30_, 97PLA/3Ti_50_Si_50_, 97PLA/3Ti_40_Si_60_, and 97PLA/3TiO_2_SiO_2_ were fully degraded at 840, 300, 420, 420, and 560 min, respectively. Moreover, the PLA composite containing 5 wt.% of Ti_70_Si_30_ degraded faster than all the composites and it was fully degraded in approximately 240 min. Consequently, it could be concluded that the rate of the hydrolytic degradation of PLA composite films can be controlled by the filler content. This result is in agreement with Buzarovka and Grozdanov (2012) [12].

### 3.8. In Vitro Degradation

The degradation of PLA in PLA composites involves several processes such as water uptake, ester cleavage and formation if there are oligomer fragments, the dissolution of the oligomer fragment, etc., [61]; as a result, factors affecting the hydrolysis tendency of PLA would control the degradation of PLA. The long-term hydrolytic degradation of PLA and PLA composite films in a phosphate buffered saline (PBS) (pH = 7.4 ± 0.2) solution at 37 °C was evaluated by mass loss in 56 days. Figure 11 illustrates the mass loss of the PLA and PLA composite with the degradation time. From 0 to 14 days, all of the samples exhibited a dramatic increase in mass loss with increasing immersion time. After this period, the mass loss of all samples accelerated gradually. PLA incorporating with 3 wt.% of TiO_2_, SiO_2_, Ti_70_Si_30_, Ti_50_Si_50_, Ti_40_Si_60_, and TiO_2_SiO_2_ exhibited higher weight loss as a function of immersion time than neat PLA. In this case, TiO_2_, SiO_2_, Ti_70_Si_30_, Ti_50_Si_50_, Ti_40_Si_60_, and TiO_2_SiO_2_ dispersed in the PLA matrix, the water molecules penetrated easier within the samples to generate the degradation process and might have be absorbed into the gap between the conglomeration of the nanoparticles due to the agglomeration of the nanofiller. Consequently, a long time is spent on diffusion into the PLA matrix. Therefore, the degradation rate increased in the first period and reached its maximum [62]. In addition, the mass loss of the PLA composite was also found to increase with an inclining amount of Ti_70_Si_30_ to 5 wt.%. Consequently, it could be concluded that the rate of the long-term degradation of the PLA composite films depended upon the content of the mixed oxide loading. This result was connected to the hydrophilicity of TiO_2_, SiO_2_, Ti_70_Si_30_, Ti_50_Si_50_, Ti_40_Si_60_, and TiO_2_SiO_2_, as well as the high-water absorption of the composites [63]. Regarding changes in the tensile strength, elongation at the break and Young’s modulus of the PLA and PLA composite films are shown in Table 5. This table shows that the tensile strength and elongation at the break of the PLA and all of the PLA composite films decreased significantly after 28 days of in vitro degradation. The result suggests that the PLA and PLA composite films were mechanically stable during 28 days of in vitro degradation. The tensile strengths of the PLA and PLA composites decreased after 28 days of degradation with microcracks appearing on part of their surfaces. It was supposed that the PLA composites would lose their mechanical strengths quickly after the microcracks developed over the whole area of the fibers [64].

### 3.9. Antimicrobial Activity

Metal oxides hold greater antibacterial efficiency, and their reinforcement in polymer composites expressively expands the antimicrobial properties of the film, which is desired in biomedical and food packaging applications. Bacteria are generally characterized by the cell membrane, which is composed mostly of a homogeneous peptidoglycan layer (which consists of amino acids and sugar). Gram-positive bacteria such as *Staphylococcus aureus* have one cytoplasm membrane with multilayers of the peptidoglycan polymer and a thicker cell wall (20–80 nm) [65], whereas in Gram-negative bacteria such as *Escherichia coli*, the bacteria wall is composed of two cell membranes, and an outer membrane and a plasma membrane with a thin layer of peptidoglycan with a thickness of 7–8 nm [65].

TiO_2_ nanoparticles are known for their antibacterial activity, and recent studies have confirmed their efficiency as antibacterial agents [65,66]. As a result, the antibacterial activity of PLA incorporated with 3 wt.% of Ti_70_Si_30_, Ti_50_Si_50_, and TiO_2_SiO_2_, and 5 wt.% of Ti_70_Si_30_, to form composites was compared to the antibacterial activity of PLA adding 3 wt.% of TiO_2_. The results of the antimicrobial activity of the Gram-negative bacteria (*Escherichia coli* or *E. coli*) and Gram-positive bacteria (*Staphylococcus aureus* or *S. aureus*) of the PLA and PLA composites are shown in Table 6 and Table 7, respectively. The number of bacteria *Escherichia coli* and bacteria *Staphylococcus aureus* on the PLA and PLA composite films at time 0 h (at dilution 10^−3^) and 24 h (at dilution 10^0^) are shown in Figure 12.

The antimicrobial activity (R) values of Neat PLA film against *E. coli* and *S. aureus* were 0.09 and 0, respectively. This result shows that PLA has no significant antibacterial effects on *E. coli* and *S. aureus*. However, the 97PLA/3TiO_2_ composite film exhibited the highest antimicrobial activity (R = 5.96 against *E. coli* and R = 4.35 against *S. aureus*). As shown in Table 6, all of PLA composites exhibited an antimicrobial activity agent against *E. coli* (if R ≥ 2 antimicrobial effectiveness). Moreover, the results confirm that PLA incorporated with 3wt.% of Ti_70_Si_30_ has sufficient antimicrobial effectiveness. Likewise, PLA with the addition of 3 wt.% of TiO_2_ and Ti_70_Si_30_ exhibited an antibacterial effect on *S. aureus*. This was due to TiO_2_ and Ti_x_Si_y_ oxide utilizing a similar mechanism against bacterial growth by directly damaging the bacterial surface. Adding TiO_2_ and Ti_x_Si_y_ oxide into PLA could lead to reduced soluble protein expression by suppressing the synthesis of nucleic acids. Thus, TiO_2_ and Ti_x_Si_y_ oxide antibacterial action against *S. aureus* was probably through inhibiting the synthesis of nucleic acid, thereby reducing protein synthesis against bacterial growth [67]. The mechanism referred to as the antimicrobial action of TiO_2_ is commonly associated with reactive oxygen species (ROS) with high oxidative potentials produced under a band-gap irradiation photo-induced charge in the presence of O_2_ [68]. Pleskova et al. investigated the bactericidal activity of the TiO_2_ film and discovered that *S. aureus* is swelled by TiO_2_ through damaging the cell membrane [69]. In addition, the increasement of Ti_70_Si_30_ loading to 5 wt.% improved the antimicrobial effectiveness of the PLA composites. However, 97PLA/3Ti_50_Si_50_ and 97PLA/3TiO_2_SiO_2_ exhibited low antimicrobial activity (R = 2.75, 3.43 against *E. coli* and R = 1.83, 1.79 against *S. aureus*). This was due to a higher SiO_2_ content in Ti_50_Si_50_ and TiO_2_SiO_2_, resulting in a lower efficiency of antimicrobial activity. In addition, the variation in the microorganism structure between the Gram-negative *(E. coli)* and Gram-positive (*S. aureus*) bacteria may explain the difference in the antibacterial effect of samples against *E. coli* and *S. aureus*. Both bacteria have similar internal, but different external structures. The peptidoglycan layer of Gram-positive bacteria is thick and includes teichoic and lipoteichoic acids. A Gram-negative bacterium has a thin peptidoglycan layer and an outer membrane made up of proteins, phospholipids, and lipopolysaccharides. Therefore, *S. aureus* needs longer contact time or higher catalyst concentrations to achieve the same effect as *E. coli* [70].

## 4. Conclusions

The aim of this study was to examine the influence of 3wt.% of TiO_2_, SiO_2_, Ti_70_Si_30_, Ti_50_Si_50_, Ti_40_Si_60_, and TiO_2_SiO_2_, and 5 wt.% of Ti_70_Si_30_ on the mechanical properties, thermal properties, morphological properties, degradation behavior, and antimicrobial activity of PLA. The PLA and PLA composites films were obtained by the solvent casting method. The addition of Ti_70_Si_30_ and Ti_50_Si_50_ into the PLA film slightly improved the tensile strength and Young’s modulus of PLA. The incorporation of 5 wt.% of Ti_70_Si_30_ was found to decrease the cold crystallization temperature and increased the degree of crystallinity of PLA. It can be concluded that Ti_70_Si_30_ nanoparticles can act as a good nucleating agents for PLA. The thermal stability of PLA was enhanced with the incorporation of TiO_2_ and SiO_2_. The water vapor transmission rate (WVTR) of PLA was significantly increased by the incorporation of SiO_2_, Ti_70_Si_30_, and Ti_50_Si_50_ nanoparticles. This is due to the hydrophilicity of the nanoparticles. In addition, efficiency of degrading MB is TiO_2_ > Ti_70_Si_30_ > TiO_2_SiO_2_ > Ti_50_Si_50_ > Ti_40_Si_60_ > SiO_2_, respectively. Moreover, the increase in Ti_70_Si_30_ loading to 5 wt.% improved the efficiency of the photocatalytic activity of PLA. All of the nanoparticles were able to remove UV light and, in particular, TiO_2_ and Ti_70_Si_30_ enhanced a stronger higher UV-shielding potential. The hydrolytic degradation and in vitro degradation of PLA are important properties of the variety of application such as biomedical application and food packaging. PLA incorporated with 3wt.% of SiO_2_, Ti_70_Si_30_, Ti_50_Si_50_, Ti_40_Si_60_, and TiO_2_SiO_2_ exhibited much higher weight loss as a function of time than neat PLA. The weight loss of the PLA composite was also found to increase with increasing Ti_70_Si_30_ to 5 wt.%. Furthermore, PLA with the addition of TiO_2_ and Ti_70_Si_30_ exhibited an excellent antibacterial effect on Gram-negative bacteria (*E. coli*) and Gram-positive bacteria (*S. aureus*), indicating the improved antimicrobial effectiveness of PLA composites.

## Figures and Tables

**Figure 1 polymers-14-03310-f001:**
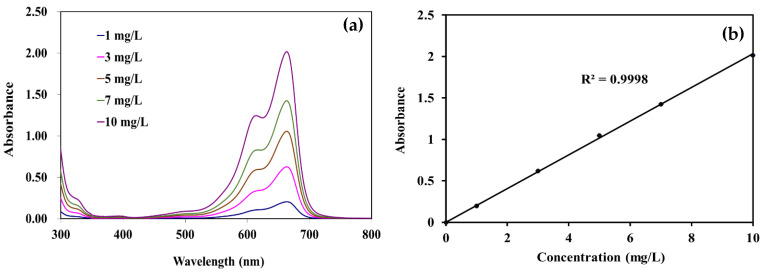
(**a**) Wavelength–absorbance curves and (**b**) calibration curve of methylene blue aqueous solutions.

**Figure 2 polymers-14-03310-f002:**
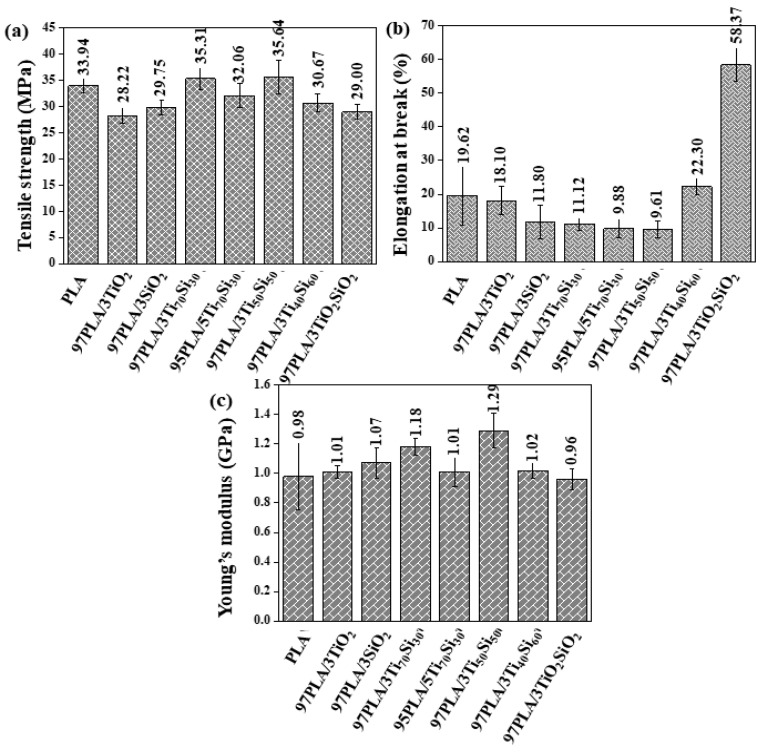
Comparison of the (**a**) Tensile strength, (**b**) Elongation at break, and (**c**) Young’s modulus of PLA, PLA/TiO_2_, PLA/SiO_2_, PLA/Ti_x_Si_y_, and PLA/TiO_2_SiO_2_ composites.

**Figure 3 polymers-14-03310-f003:**
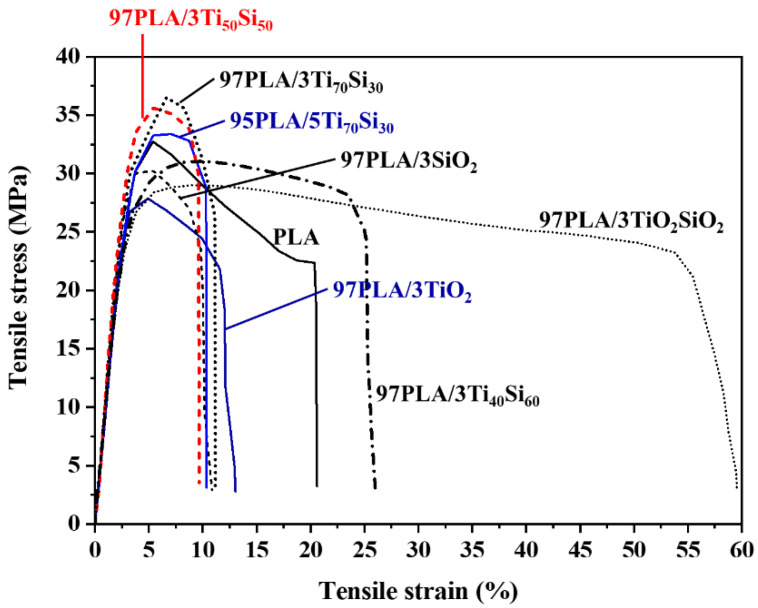
Stress–strain curves of PLA, PLA/TiO_2_, PLA/SiO_2_, PLA/Ti_x_Si_y_, and PLA/TiO_2_SiO_2_ composites.

**Figure 4 polymers-14-03310-f004:**
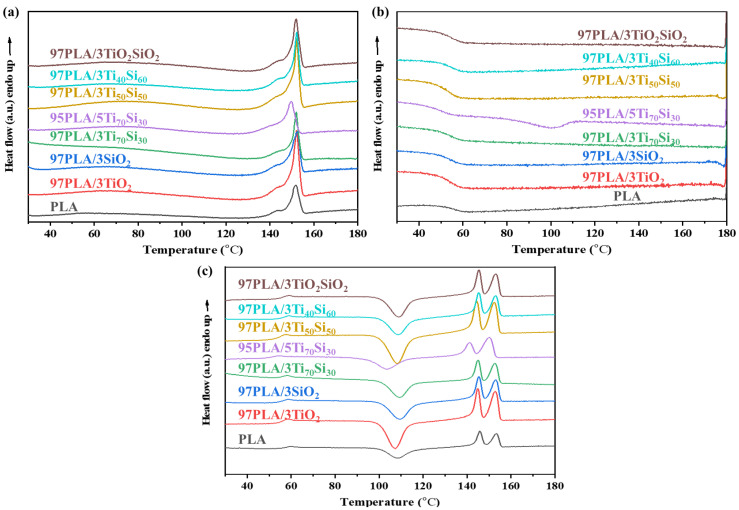
DSC thermogram of (**a**) first heating, heating rate 5 °C/min, (**b**) cooling, cooling rate 5 °C/min, (**c**) second heating, heating rate 5 °C/min of PLA, PLA/TiO_2_, PLA/SiO_2_, PLA/Ti_x_Si_y_, and PLA/TiO_2_SiO_2_ composites.

**Figure 5 polymers-14-03310-f005:**
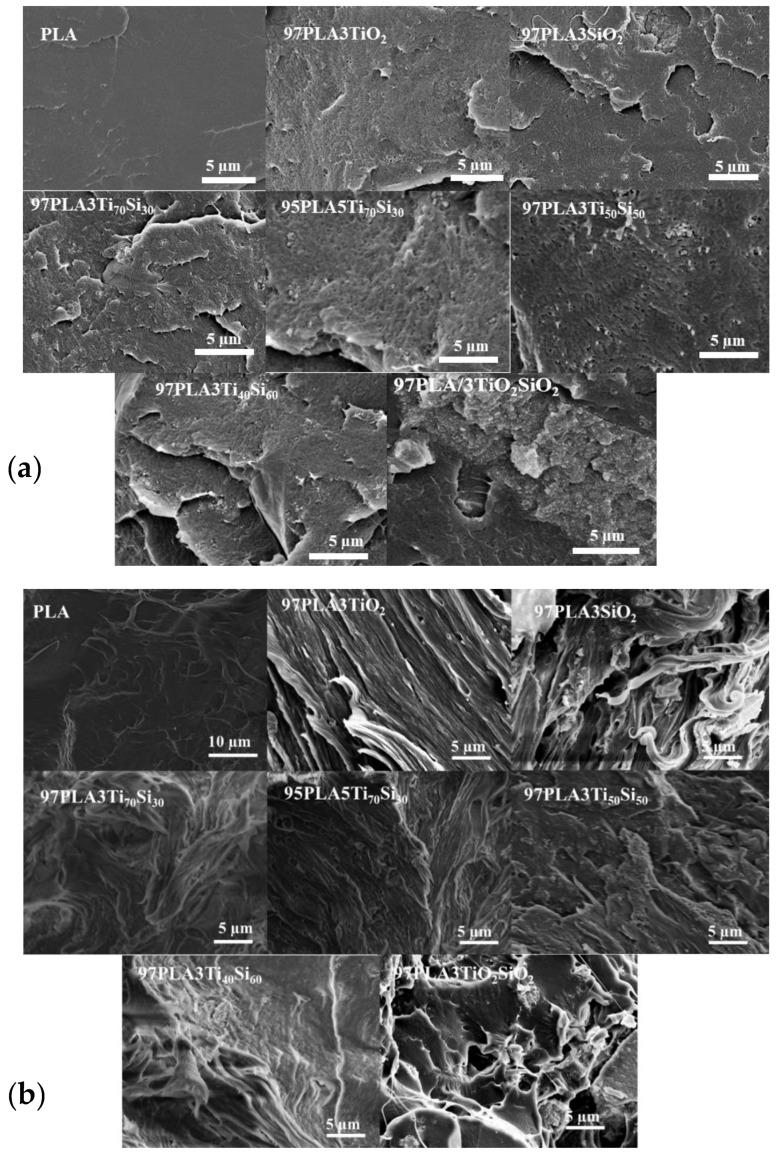
SEM micrographs (×2.5 k, WD = 13–15 mm, acceleration voltage 9–12 kV) of (**a**) the fracture surface and (**b**) after tensile testing of PLA, PLA/TiO_2_, PLA/SiO_2_, PLA/Ti_x_Si_y_, and PLA/TiO_2_SiO_2_ composites.

**Figure 6 polymers-14-03310-f006:**
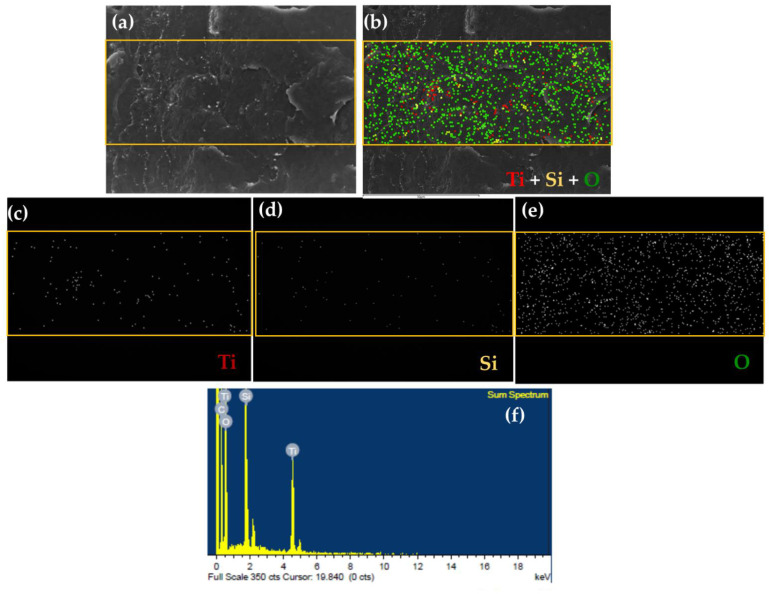
(**a**) SEM images (**b**) over all elements in the PLA/Ti_70_Si_30_ composite and the corresponding elemental mapping analysis of (**c**) Ti, (**d**) Si, (**e**) O, and (**f**) EDX spectra.

**Figure 7 polymers-14-03310-f007:**
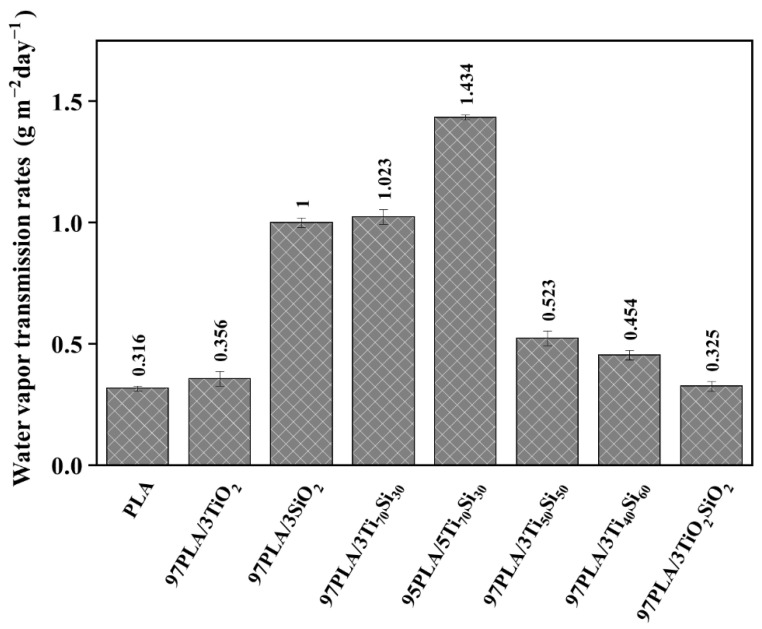
Water vapor transmission rate (WVTR) of PLA, PLA/TiO_2_, PLA/SiO_2_, PLA/Ti_x_Si_y_, and PLA/TiO_2_SiO_2_ composites.

**Figure 8 polymers-14-03310-f008:**
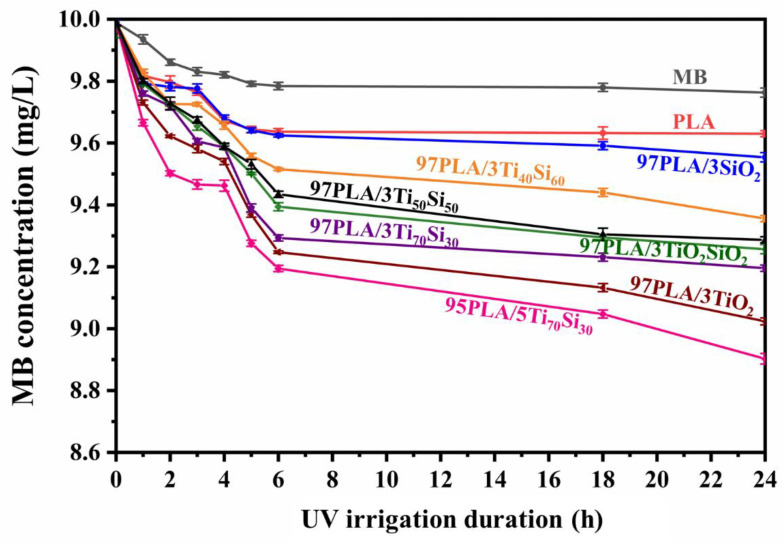
Concentration of methylene blue (MB) due to absorption of PLA, PLA/TiO_2_, PLA/SiO_2_, PLA/Ti_x_Si_y_, and PLA/TiO_2_SiO_2_ composite films under UV irrigation.

**Figure 9 polymers-14-03310-f009:**
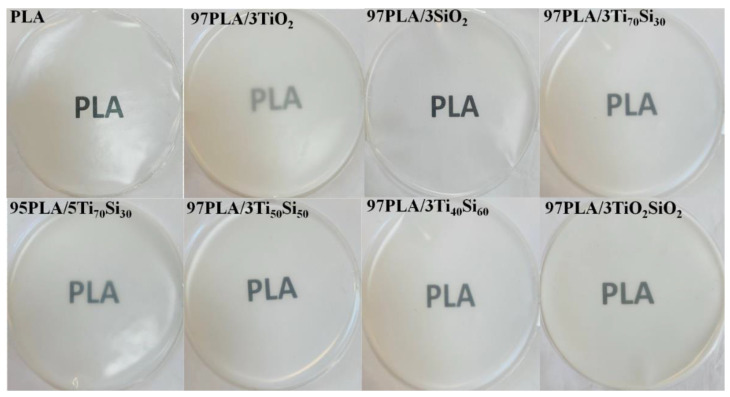
Photographs of films prepared from PLA, PLA/TiO_2_, PLA/SiO_2_, PLA/Ti_x_Si_y_, and PLA/TiO_2_SiO_2_ composites (250 ± 4.68 µm thickness).

**Figure 10 polymers-14-03310-f010:**
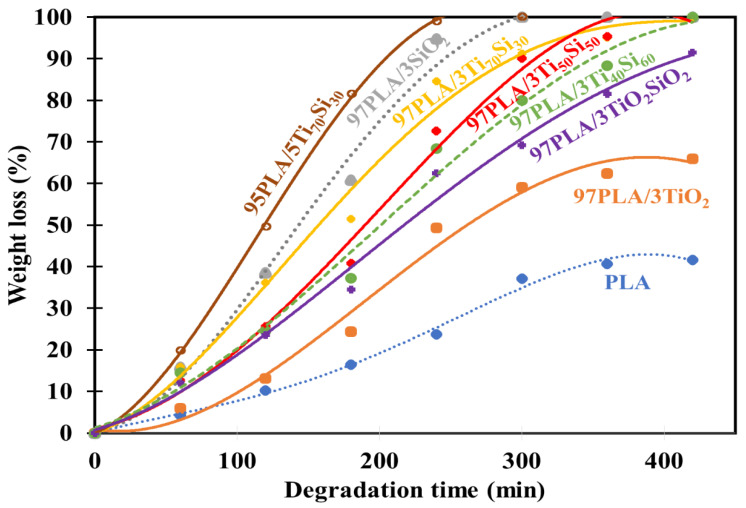
Weight loss of hydrolytic degradation of PLA, PLA/TiO_2_, PLA/SiO_2_, and PLA/Ti_x_Si_y_ composite films as functions of degradation time.

**Figure 11 polymers-14-03310-f011:**
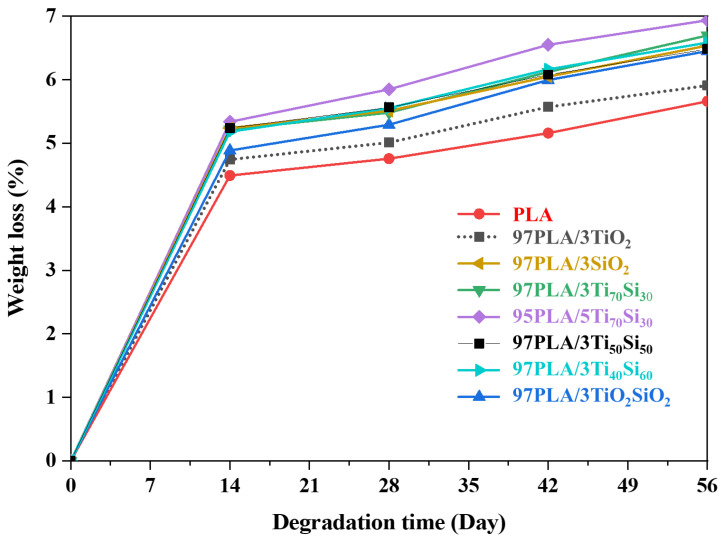
Weight loss of PLA, PLA/TiO_2_, PLA/SiO_2_, PLA/Ti_x_Si_y_, and PLA/TiO_2_SiO_2_ composites films after different periods of in vitro degradation.

**Figure 12 polymers-14-03310-f012:**
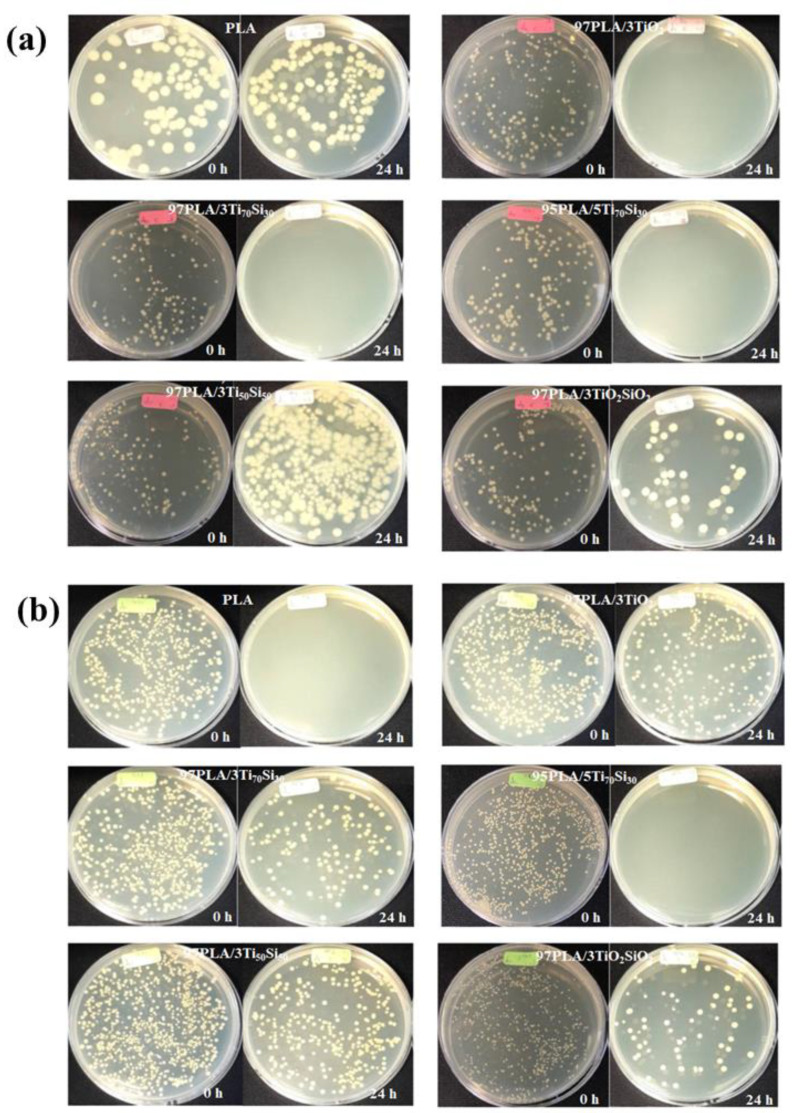
The number of (**a**) bacteria *Escherichia coli* and (**b**) bacteria *Staphylococcus aureus* on PLA and PLA composite films at time 0 h (at dilution 10^−3^) and 24 h (at dilution 10^0^).

**Table 1 polymers-14-03310-t001:** Tensile properties of PLA, PLA/TiO_2_, PLA/SiO_2_, PLA/Ti_x_Si_y_, and PLA/TiO_2_SiO_2_ composites.

Sample	Tensile Strength (MPa)	Elongation at Break (%)	Young’s Modulus (GPa)
**PLA**	33.94 ± 1.38	19.62 ± 8.64	0.98 ± 0.23
**97PLA/3TiO_2_**	28.22 ± 1.47	18.1 ± 4.24	1.01 ± 0.04
**97PLA/3SiO_2_**	29.75 ± 1.38	11.8 ± 4.96	1.07 ± 0.10
**97PLA/3Ti_70_Si_30_**	35.64 ± 3.21	11.12 ± 1.81	1.18 ± 0.06
**95PLA/5Ti_70_Si_30_**	32.06 ± 2.29	9.88 ± 2.81	1.01 ± 0.10
**97PLA/3Ti_50_Si_50_**	35.31 ± 2.01	9.61 ± 2.57	1.29 ± 0.12
**97PLA/3Ti_40_Si_60_**	30.67 ± 1.74	22.30 ± 2.53	1.02 ± 0.05
**97PLA/3TiO_2_SiO_2_**	29.00 ± 1.42	58.37 ± 5.00	0.96 ± 0.07

**Table 2 polymers-14-03310-t002:** Thermal characteristics of PLA, PLA/TiO_2_, PLA/SiO_2_, PLA/Ti_x_Si_y_, and PLA/TiO_2_SiO_2_ composites (the second heating, heating rate 5 °C/min).

Samples	T_g_, °C	T_cc_, °C	ΔH_c_, J/g	T_m1_, °C	T_m2_, °C	ΔH_m_, J/g	X_c_,%
**PLA**	58.87	109.47	36.15	145.84	153.35	31.73	33.86
**97PLA/3TiO_2_**	56.86	107.36	33.61	144.89	152.72	33.17	36.50
**97PLA/3SiO_2_**	56.64	108.22	33.06	145.39	145.33	33.31	36.65
**97PLA/3Ti_70_Si_30_**	54.81	107.3	34.41	145.07	152.75	32.71	35.99
**95PLA/5Ti_70_Si_30_**	51.84	103.09	26.80	141.29	150.22	32.18	36.15
**97PLA/3Ti_50_Si_50_**	55.96	107.14	33.06	144.41	152.50	34.44	37.89
**97PLA/3Ti_40_Si_60_**	57.40	107.72	34.16	145.33	153.01	33.38	36.73
**97PLA/3TiO_2_SiO_2_**	57.46	107.79	30.54	145.39	153.07	32.31	35.55

**Table 3 polymers-14-03310-t003:** Thermal degradation temperature of PLA, PLA/TiO_2_, PLA/SiO_2_, PLA/Ti_x_Si_y_, and PLA/TiO_2_SiO_2_ composites.

Samples	T_0__.__05_, °C	T_0__.__5_, °C	T_d,_ °C	T_f_, °C	Residual, %
**PLA**	321.67	360.33	358.83	424.64	1.20
**97PLA/3TiO_2_**	336.17	363.67	363.17	429.22	4.07
**97PLA/3SiO_2_**	331.17	361.17	359.33	427.27	4.14
**97PLA/3Ti_70_Si_30_**	304.00	351.83	350.17	408.68	3.91
**95PLA/5Ti_70_Si_30_**	284.50	347.33	346.33	407.16	5.46
**97PLA/3Ti_50_Si_50_**	324.50	358.36	356.50	425.49	4.22
**97PLA/3Ti_40_Si_60_**	326.50	358.67	356.83	416.83	3.53
**97PLA/3TiO_2_SiO_2_**	330.67	362.33	361.33	418.69	4.50

**Table 4 polymers-14-03310-t004:** Transmittance (%) and opacity values of PLA, PLA/TiO_2_, PLA/SiO_2_, PLA/Ti_x_Si_y_, and PLA/TiO_2_SiO_2_ composite films in the visible, UV-A, UV-B, and UV-C regions.

Sample	Transmittance, %				Opacity (AU.nm∙mm^−1^)
	UV-C (240 nm)	UV-B (300 nm)	UV-A (360 nm)	Visible (600 nm)	
**PLA**	1.24	24.47	34.77	51.39	1.16
**97PLA/3TiO_2_**	0.00	0.12	0.41	1.98	6.81
**97 PLA/3SiO_2_**	0.04	9.48	18.03	38.37	2.31
**97 PLA/3Ti70Si30**	0.00	0.28	0.75	14.00	3.42
**95 PLA/5Ti70Si30**	0.00	0.00	0.36	4.06	6.33
**97 PLA/3Ti50Si50**	0.00	0.32	2.37	15.65	3.22
**97 PLA/3Ti40Si60**	0.00	0.03	0.81	10.48	3.92
**97 PLA/3TiO_2_SiO_2_**	0.00	0.19	1.43	7.30	4.74

**Table 5 polymers-14-03310-t005:** Tensile properties of properties of PLA, PLA/TiO_2_, PLA/SiO_2_, and PLA/Ti_x_Si_y_ composite films after different period of in vitro degradation.

Time (Day)	Sample	Tensile Strength (MPa)	Elongation at Break (%)	Young’s Modulus (GPa)
**0**	PLA	33.94 ± 1.38	19.62 ± 8.64	0.98 ± 0.23
	97PLA/3TiO_2_	28.22 ± 1.47	18.1 ± 4.24	1.01 ± 0.04
	97PLA/3SiO_2_	29.75 ± 1.38	11.8 ± 4.96	1.07 ± 0.10
	97PLA/3Ti_70_Si_30_	35.64 ± 3.21	11.12 ± 1.81	1.18 ± 0.06
	95PLA/5Ti_70_Si_30_	32.06 ± 2.29	9.88 ± 2.81	1.01 ± 0.10
	97PLA/3Ti_50_Si_50_	35.31 ± 2.01	9.61 ± 2.57	1.29 ± 0.12
	97PLA/3Ti_40_Si_60_	30.67 ± 1.74	22.30 ± 2.53	1.02 ± 0.05
	97PLA/3TiO_2_SiO_2_	29.00 ± 1.42	58.37 ± 5.00	0.96 ± 0.07
**14**	PLA	29.43 ± 3.02	3.81 ± 1.61	1.51 ± 0.14
	97PLA/3TiO_2_	23.05 ± 2.87	2.58 ± 0.54	1.46 ± 0.31
	97PLA/3SiO_2_	23.53 ± 2.07	2.78 ± 0.27	1.39 ± 0.40
	97PLA/3Ti_70_Si_30_	33.56 ± 3.45	3.69 ± 0.61	1.74 ± 0.14
	95PLA/5Ti_70_Si_30_	31.75 ± 2.31	3.55 ± 0.70	1.49 ± 0.03
	97PLA/3Ti_50_Si_50_	30.75 ± 2.51	3.45 ± 0.80	1.29 ± 0.04
	97PLA/3Ti_40_Si_60_	27.99 ± 1.18	4.46 ± 0.95	1.21 ± 0.44
	97PLA/3TiO_2_SiO_2_	24.83 ± 2.35	3.25 ± 0.38	1.44 ± 0.56
**28**	PLA	19.95 ± 2.45	2.06 ± 0.38	1.75 ± 0.25
	97PLA/3TiO_2_	18.69 ± 2.02	1.98 ± 0.23	2.02 ± 0.00
	97PLA/3SiO_2_	n/a *	n/a	n/a
	97PLA/3Ti_70_Si_30_	29.35 ± 2.07	2.56 ± 0.12	1.91 ± 0.39
	95PLA/5Ti_70_Si_30_	10.03 ± 2.40	1.11 ± 0.33	n/a
	97PLA/3Ti_50_Si_50_	18.03 ± 2.80	2.11 ± 0.23	n/a
	97PLA/3Ti_40_Si_60_	17.56 ± 2.01	2.08 ± 0.13	n/a
	97PLA/3TiO_2_SiO_2_	17.39 ± 2.46	1.79 ± 0.35	n/a
**42**	PLA	6.95 ± 2.49	1.33 ± 0.19	n/a
	97PLA/3TiO_2_	0.26 ± 0.11	0.81 ± 0.10	n/a
	97PLA/3SiO_2_	n/a	n/a	n/a
	97PLA/3Ti_70_Si_30_	15.97 ± 2.62	1.52 ± 0.11	n/a
	95PLA/5Ti_70_Si_30_	n/a	n/a	n/a
	97PLA/3Ti_50_Si_50_	5.97 ± 2.63	0.62 ± 0.52	n/a
	97PLA/3Ti_40_Si_60_	n/a	n/a	n/a
	97PLA/3TiO_2_SiO_2_	3.24 ± 1.79	1.12 ± 0.64	n/a

* n/a = not available.

**Table 6 polymers-14-03310-t006:** Antimicrobial activity of Gram-negative bacteria (*Escherichia coli*) of PLA, PLA/TiO_2_, PLA/Ti_x_Si_y_, and PLA/TiO_2_SiO_2_ composites.

Samples	Blank (Ut) (*t* = 24 h)	Sample (At) (*t* = 24 h)	Antimicrobial Activity ^a^ (R)
	Log CFU/mL	Log CFU/mL	
**PLA**	5.96 ± 0.01	5.87 ± 0.02	0.09
**97PLA/3TiO_2_**	5.96 ± 0.01	0.00 ± 0.00	5.96
**97PLA/3Ti_70_Si_30_**	5.96 ± 0.01	0.00 ± 0.00	5.96
**95PLA/5Ti_70_Si_30_**	5.96 ± 0.01	0.00 ± 0.00	5.96
**97PLA/3Ti_50_Si_50_**	5.96 ± 0.01	3.21 ± 0.04	2.75
**97PLA/3TiO_2_SiO_2_**	5.96 ± 0.01	2.53 ± 0.04	3.43

^a^ Antibacterial activity (R) ≥ 2 = antimicrobial effectiveness.

**Table 7 polymers-14-03310-t007:** Antimicrobial activity of Gram-positive bacteria (*Staphylococcus aureus*) of PLA, PLA/TiO_2_, PLA/Ti_x_Si_y_, and PLA/TiO_2_SiO_2_ composites.

Samples	Blank (Ut) (*t* = 24 h)	Sample (At) (*t* = 24 h)	Antimicrobial Activity ^a^ (R)
	Log CFU/mL	Log CFU/mL	
**PLA**	4.35 ± 0.04	4.35 ± 0.04	0
**PLA/3TiO_2_**	4.35 ± 0.04	0.00 ± 0.00	4.35
**PLA/3Ti_70_Si_30_**	4.35 ± 0.04	2.96 ± 0.01	2.04
**PLA/5Ti_70_Si_30_**	4.35 ± 0.04	0.00 ± 0.00	4.35
**PLA/3Ti_50_Si_50_**	4.35 ± 0.04	3.17 ± 0.08	1.83
**PLA/3TiO_2_SiO_2_**	4.35 ± 0.04	3.21 ± 0.01	1.79

^a^ Antibacterial activity (R) ≥ 2 = antimicrobial effectiveness.

## Data Availability

Not applicable.

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
