# Peer review of "Effects of Titanium–Silica Oxide on Degradation Behavior and Antimicrobial Activity of Poly (Lactic Acid) Composites"

_polymers, 2022, doi:10.3390/polym14163310_

Round 1
Reviewer 1 Report
The article, "Effects of titanium-silica oxide on degradation behavior and antimicrobial activity of Poly (lactic acid) composites," presents a fascinating look at preparing a new type of material with various applications. The results are fascinating and worthy of publication, but with minor adjustments:
- On what basis was equation 2.1 developed? Did your group develop it, or was it modified from an existing one? Why was it used in this form?
- Why was the absorption at 600nm used in equation 2.3? In the text was a different value mentioned.
- Why were such TixSix combinations chosen, and why are the results for 95PLA/5Ti50Si50 and 95PLA/5Ti40Si60 and others not shown? There is no clear explanation of the choice of ratios
- SEM-EDS for individual elements showing the spatial distribution of Ti and Si relative to PLA is missing. These results would show the degree of aggregation and the percentage of Ti and Si in the sample. These results would be crucial in discussing the results as was done in the article in the current version.
- Why were variable-sized nanoparticles chosen? These results are an additional factor that can modify the material's properties, bypassing the effect of the TixSix combination. It would be helpful to take nanoparticles of equal or similar size and check the properties of the composite. In the current form, it is not possible to meaningfully compare the data and samples with each other.
- Why do 97PLA/3Ti50Si50 and 97PLA/3TiO2SiO2 show low biological response (Tables 5 and 6)?
- Why does PLA inhibit the growth of S. aureus bacteria after 24h while composite systems do not?
- There is no explanation of the mechanisms of the antibacterial effect of the composites relative to pure PLA, especially in the context of such strange results as shown in Tables 5 and 6 and Figure 17.
- Names of bacteria should be written in italics,
- Wrong caption on the x-axis in Fig. 2. The caption for this figure could also be refined.
- line 279: SEM is mentioned, but there is no reference where this data is located
- line 618: how was the specific surface area determined? There is no information about the method of measurement and comparison of values between samples
- line 620-630: were the films of equal thickness?
Reviewer 2 Report
Manuscript ID: polymers-1840281
Title: Effects of titanium-silica oxide on degradation behavior and antimicrobial activity of Poly (lactic acid) composites
In this work, the authors examined the influence of 3wt.% of TiO2, SiO2, Ti70Si30, Ti50Si50, Ti40Si60, TiO2SiO2 and 5wt.% of Ti70Si30 on mechanical properties, photocatalytic efficiency, antibacterial property, permeability tests, and biodegradability of PLA. The effect of using TixSiy oxides as a filler in PLA composites on these properties was compared with the use of TiO2, SiO2 and TiO2SiO2. Also, the PLA composite films produced with TixSiy oxide were transparent, capable of screening UV radiation and exhibited superior antibacterial efficacy, making them an excellent food packaging material. Overall, the manuscript is well structured and the topic can be suitable for in this journal. However, a major revision is suggested for the current manuscript due to the following concerns:
1. The language of this manuscript need further polishing and improvement. Please check the main text and remove the typos before the re-submission.
2. The manuscript is too long and there are too many figures in the main text. Please show part of them in the supporting information. Additionally, some figures can be deleted such as Figure 1.
3. The value of Absorbance in Fig 3 is wrong and not consistent with that in Fig 2. Please check it.
4. Actually, Fig 2 and Fig 3 can be shown in one figure.
5. From Figure 7, it can be seen that the presence of TiO2, SiO2 and titanium-silica oxide has changed the thermal decomposition behavior of PLA, due to the different curves of the samples. Please explain it.
6. As for photocatalytic activity of PLA composite films, the dosage of PLA composite in the MB dye solutions should be added.
7. As for the Morphological properties, the existence of titanium-silica oxide in the composite film can be further confirmed by EDS mapping analysis.
8. In the figure 12, the error bars should be added and meanwhile the discussion need further improvement. The authors can refer to the literature: https://doi.org/10.1016/j.cej.2020.125347; https://doi.org/10.1007/s10854-019-00932-x.
9.How about the stability of the PLA composite for the long-term photocatalytic reaction? Please add the materials characterizations of the PLA materials after photocatalytic test.
Round 2
Reviewer 1 Report
Thank you for the responses.
Reviewer 2 Report
The previous comments have been well addressed and this version can be considered for publication.